# Neonatal Extracellular Superoxide Dismutase Knockout Mice Increase Total Superoxide Dismutase Activity and VEGF Expression after Chronic Hyperoxia

**DOI:** 10.3390/antiox10081236

**Published:** 2021-08-01

**Authors:** Maxwell Mathias, Joann Taylor, Elizabeth Mendralla, Marta Perez

**Affiliations:** 1Division of Neonatology, Department of Pediatrics, Northwestern University Feinberg School of Medicine, Chicago, IL 60611, USA; j-hinz@northwestern.edu (J.T.); emendral@student.touro.edu (E.M.); mtperez@luriechildrens.org (M.P.); 2Ann & Robert H. Lurie Children’s Hospital of Chicago, Chicago, IL 60611, USA

**Keywords:** extracellular superoxide dismutase, bronchopulmonary dysplasia, vascular endothelial growth factor, nitric oxide signaling

## Abstract

Bronchopulmonary dysplasia (BPD) is a common lung disease affecting premature infants that develops after exposure to supplemental oxygen and reactive oxygen intermediates. Extracellular superoxide dismutase (SOD3) is an enzyme that processes superoxide radicals and has been shown to facilitate vascular endothelial growth factor (VEGF) and nitric oxide (NO) signaling in vascular endothelium. We utilized a mouse model of neonatal hyperoxic lung injury and SOD3 knockout (KO) mice to evaluate its function during chronic hyperoxia exposure. Wild-type age-matched neonatal C57Bl/6 (WT) and SOD3^−/−^ (KO) mice were placed in normoxia (21% FiO_2_, RA) or chronic hyperoxia (75% FiO_2_, O_2_) within 24 h of birth for 14 days continuously and then euthanized. Lungs were harvested for histologic evaluation, as well as comparison of antioxidant enzyme expression, SOD activity, VEGF expression, and portions of the NO signaling pathway. Surprisingly, KO-O_2_ mice survived without additional alveolar simplification, microvascular remodeling, or nuclear oxidation when compared to WT-O_2_ mice. KO-O_2_ mice had increased total SOD activity and increased VEGF expression when compared to WT-O_2_ mice. No genotype differences were noted in intracellular antioxidant enzyme expression or the NO signaling pathway. These results demonstrate that SOD3 KO mice can survive prolonged hyperoxia without exacerbation of alveolar or vascular phenotype.

## 1. Introduction

Bronchopulmonary dysplasia (BPD) is a common lung disease of prematurity, affecting more than 10,000 infants in the United States annually and up to 60% of infants born under 28 weeks [1,2]. BPD is a consequence of lung immaturity combined with the side-effects of life-sustaining therapeutic interventions, including supplemental oxygen (hyperoxia), which result in injury to the developing lung [3]. Hyperoxia is hypothesized to induce injury through the production of excess reactive oxygen intermediates (ROIs) [4]. Preterm infants are particularly susceptible to injury from ROI because of the immaturity of enzymatic antioxidant systems and relative deficiency of nonenzymatic antioxidants relative to term infants [5,6,7,8,9]. Because of the hypothesized causal pathway of hyperoxia–ROI–lung injury, there have been multiple clinical trials of administration of either antioxidant enzymes or nonenzymatic antioxidants; however, to date, none have demonstrated superiority over standard care or have received approval from the FDA [10,11,12,13,14]. Since these trials were conducted, our knowledge of cell-specific and compartment-specific redox signaling has vastly expanded, and it is clear that therapeutic intervention for neonatal hyperoxic lung injury (HLI) will require a more granular understanding of antioxidant system function during hyperoxia exposure [15].

Extracellular superoxide dismutase (SOD3) is an antioxidant enzyme found in high concentration in the lungs, and it is 95% tissue-bound at the interface between pulmonary vascular smooth muscle and pulmonary vascular endothelium [16,17]. SOD3 converts superoxide radical to hydrogen peroxide (H_2_O_2_), which can cross lipid bilayers and be converted to water by numerous intracellular antioxidant enzymes or can modify cysteine thiols on proteins for redox-dependent signaling [18]. SOD3 plays an important role in the development of the newborn lung. Its expression increases fivefold between weeks 1 and 3 of life and sevenfold between weeks 1 and 8 [5]. SOD3 knockout (KO) mice have simplified alveoli, pulmonary vascular remodeling, and increased susceptibility to bleomycin-induced lung injury at 3 and 4 weeks of life in comparison to controls [19,20,21]. In contrast, SOD3 overexpression protects against HLI in newborn animals, but does not appear to have any effect on alveolar or pulmonary vascular development during normoxia (21% FiO_2_) [22,23,24]. The specific function of SOD3 in neonatal lung development and how this function is altered by hyperoxia exposure remains unknown.

One hypothesized function of SOD3 is that it protects nitric oxide (NO) signaling, whose function is critical to lung and pulmonary vascular development [24,25,26,27,28]. In the pulmonary vascular endothelium, NO is produced by the endothelial nitric oxide synthase (eNOS), diffuses into vascular smooth muscle, and binds soluble guanylate cyclase (sGC). Activated sGC catalyzes the conversion of GTP to cGMP, an important second messenger in vascular smooth muscle function and lung growth. Each step of this pathway has been shown to play an important role in early lung development, and sGC depletion has been shown to increase susceptibility to HLI [29,30,31]. SOD3 may protect NO bioavailability by reducing the extracellular concentration of superoxide, which can attack NO to form peroxynitrite (ONOO^−^), itself a highly reactive and toxic intermediate [25,32,33]. In addition to direct effects on NO bioavailability, SOD3 can enhance NO signaling by increasing NO production in the vascular endothelium through its effect on vascular endothelial growth factor (VEGF) signaling. SOD3-derived H_2_O_2_ has been shown to potentiate the effect of VEGF on VEGF receptor 2 (VEGFR-2) phosphorylation [34]. VEGFR-2 phosphorylation induces the expression and activation of the endothelial nitric oxide synthase (eNOS), providing a link between the effects of SOD3 on VEGF and the NO signaling pathway [35,36].

In the present study, we evaluated the combined effects of SOD3 KO and chronic hyperoxia (O_2_) on lung and pulmonary vascular development, and we tested the hypothesis that SOD3 protects lung development through VEGF-mediated effects on NO signaling. To our knowledge, this is the first study to investigate the effects of SOD3 KO on neonatal adaptations to chronic hyperoxia.

## 2. Materials and Methods

### 2.1. Animal Model

This study was approved by the Northwestern University Institutional Animal Care and Use Committee. SOD3^−/−^ mice (The Jackson Laboratory, Bar Harbor, ME, USA) were bred on a background of C57Bl/6N mice (Charles River, Wilmington, MA, USA) and back-crossed through multiple generations. Heterozygotes were then crossed to produce SOD3^−/−^ (KO) and SOD3^+/+^ (WT) offspring. Cages were checked daily for litters. New litters were placed in either room air (RA, 21% FiO_2_) or a plexiglass chamber (Biospherix, Lacona, NY, USA) at 75% FiO_2_ (Hyperoxia, or O_2_) from birth to P14 continuously. Dams were rotated daily to prevent oxygen toxicity. Mice were euthanized at 14 days.

### 2.2. Tissue Collection and Processing

For histology, morphometry, and immunofluorescence, the chest was exposed, and lungs were removed en bloc and inflated to 25 cm H_2_O with 10% formalin solution, fixed for 24 h, and paraffin-embedded. Blocks were cut into 5 μm thick sections and stained with hematoxylin and eosin (H&E) for basic morphometry by the Northwestern University Mouse Histology and Phenotyping Laboratory and the Stanley Manne Children’s Research Institute Histology Laboratory. For Western blotting, enzyme activity assays, and immunoprecipitation, the RV was punctured, and the pulmonary artery was flushed with 3 mL of PBS to remove blood; then, the lung tissue was removed and snap-frozen in liquid nitrogen. Samples were stored at −80 °C until they were analyzed.

### 2.3. Lung Histology

H&E slides were blinded to the researcher. For alveolar area, 4–8 nonoverlapping images of the lungs were taken at 20× magnification using an Olympus BX40 microscope (Olympus America, Melville, NY, USA) with a Moticam Pro 252A microscope camera (Motic, Kowloon, Hong Kong). Alveolar area was calculated using ImagePro 9.2 (Media Cybernetics, Rockland, MD, USA). Obvious airways and blood vessels were excluded, as were alveoli without complete borders. For pulmonary artery smooth muscle wall thickness index (MWT), 4–8 distinct resistance pulmonary arteries adjacent to airways measuring less than 100 μm diameter were photographed per slide at 40× magnification. Inner and outer borders were traced in triplicate using ImageJ Software (ImageJ, NIH, Bethesda, MD, USA) to calculate circumference, and the ratio of smooth muscle to total area was compared between groups as described previously [37,38].

### 2.4. Immunofluorescence

Unstained 5 μm thick sections were obtained from paraffin-embedded lung samples. For von Willebrand Factor (vWF) immunofluorescence, slides were heated to 60 °C for 10 min, and then cleaned and deparrafinized by serial dilutions of ethanol. Samples were then incubated in proteinase K solution (Sigma-Aldrich, St. Louis, MO, USA) for antigen retrieval at 37 °C for 15 min. Samples were blocked with 5% bovine serum albumin (BSA; Sigma-Aldrich) in Tris-buffered saline containing 0.1% Tween-20 (1× TBST) at room temperature for 30 min, and then probed with von Willebrand Factor antibody (Agilent, Santa Clara, CA, USA; 1:50 in 5% BSA) overnight in a dark chamber. Slides were then washed in PBS twice and probed with Rhodamine Red conjugated goat anti-rabbit antibody (ThermoFisher, Waltham, MA, USA; 1:100 in 5% BSA) for 1 h in a dark chamber. Slides were again washed twice in PBS, blotted dry, and mounted prior to storage at 4 °C. Slides were imaged at 10× magnification on a NIKON Eclipse TE300 fluorescent microscope and captured using a Photometrics SenSys KAF0400 G-2 camera. Eight nonoverlapping images per slide were obtained for vessel counts. Incompletely stained vessels or vessels with diameter >100 μm were excluded.

8-Hydroxydeoxyguanosine (8-OHdG) is a product of DNA oxidation. To measure intracellular oxidative stress, 8-OHdG was quantified using immunofluorescence. Samples were deparaffinized on an automated platform (Leica Autostainer XL) and incubated in Sodium Citrate solution (pH 6) at 110 °C for 10 min in a pressure cooker for antigen retrieval. Slides were then incubated in 8-OHdG antibody (Sigma-Aldrich, 1:100) overnight at 4 °C, washed, incubated in species-specific secondary antibody (Jackson ImmunoResearch, West Grove, PA, USA; 1:500), and counterstained with DAPI. Slides were imaged at 40× magnification on a KEYENCE BZ-X fluorescent microscope (KEYENCE America, Itasca, IL, USA). A total of 4–8 nonoverlapping regions were selected, and duplicate images were obtained on blue (DAPI) and green (8-OHdG) channels. Images were converted to grayscale and analyzed using ImageJ. Nuclear borders were delineated by applying the same grayscale intensity threshold to all the DAPI images. These borders were overlaid on the 8-OHdG images, and the average 8-OHdG intensity of nuclear regions was calculated to filter out non-nuclear background fluorescence.

### 2.5. Western Blotting

Frozen lung tissue was crushed, suspended in protein lysis buffer (EMD Millipore Mg^2+^) with phosphatase inhibitor and protease inhibitor (EMD Millipore and Sigma-Aldrich), and sonicated. Protein concentration was calculated using the Bradford method [39]. A total of 40 μg of protein lysate per sample was run on 4–20% gel (ThermoFischer) and transferred to a nitrocellulose membrane (GE Life Sciences Amersham). Membranes were blocked with 5% BSA in TBST and incubated overnight with primary antibodies at 4 °C on a rocker. Membranes were washed and incubated with horseradish peroxidase (HRP)-conjugated, species-specific secondary antibodies, and again washed three times prior to imaging. SuperSignal West Femto Maximum Sensitivity Substrate (ThermoFischer Scientific, Rockford, IL, USA) was added, and chemiluminescence was detected using a molecular imager (BioRad, Hercules, CA, USA). Bands were analyzed using Image Lab (BioRad, Hercules, CA, USA), normalized to β-actin, and shown as fold difference relative to controls.

Primary antibodies were diluted in 5% BSA unless otherwise specified. Primary antibodies and concentrations used were as follows: SOD3 goat polyclonal antibody (pAb) at 1:1000 (R + D), cytosolic superoxide dismutase (SOD1) rabbit pAb at 1:1000 (Enzo P07632), mitochondrial superoxide dismutase (SOD2) rabbit pAb at 1:750 (Enzo P04179), glutathione peroxidase 1 (GPX-1) rabbit pAb at 1:1000 (Abnova 5038), catalase rabbit pAb at 1:2000 (Abcam ab52477), VEGF-A mouse monoclonal antibody (mAb) at 1:500 (Santa Cruz), sGC-α rabbit pAb at 1:2000 (Abcam), sGC-β rabbit pAb at 1:1000 (Cayman Chemical), endothelial nitric oxide synthase (eNOS) mouse mAb at 1:500 (Cayman Chemical), and β-actin rabbit mAb at 1:5000 (Cell Signaling). HRP-conjugated secondary antibodies and concentrations were as follows: goat anti-rabbit at 1:2000 (Cell Signaling), horse anti-mouse at 1:2000 (Cell Signaling), and donkey anti-goat at 1:2000 (Santa Cruz).

### 2.6. SOD Activity Assay

Protein lysates were obtained as described above. SOD enzyme activity was measured as described previously using a commercially available SOD Activity Kit (Enzo) [40]. Briefly, whole lung tissue lysates were passed through 0.45 μm filter, and protein concentration was measured. A total of 150 μg of lung protein was diluted to 75 μL (2 μg/μL) and incubated in SOD buffer, xanthine oxidase, and Water-Soluble Tetrazolium Salt 1 (WST-1) as per the manufacturer’s instructions. Xanthine was added to initiate superoxide formation, which was measured as the rate of reduction of WST-1 to WST-1 formazan, which absorbs light at 450 nm. Absorbance readings were taken at 1 min intervals until WST-1 formazan formation ceased. SOD activity was calculated as rate of inhibition of WST-1 formazan formation. These were plotted against standard SOD enzyme concentrations (0.1 U/25 μL to 10 U/25 μL) on a logarithmic scale to estimate total SOD activity.

### 2.7. sGC Activity Assay

Protein lysates were obtained as described above. The cGMP ELISA kit was purchased from Cayman Chemical. A total of 30 μg of protein was diluted to 100 μL in incubation buffer consisting of 50 mM Tris-HCl (pH 7.5, Fisher Scientific), 4 mM MgCl_2_ (Fisher Scientific), 0.5 mM 3-isobutyl-1-methylxanthine (Enzo), 7.5 mM creatine phosphate (Sigma-Aldrich), 0.2 mg/mL creatine phosphokinase (Sigma-Aldrich), 10 mM sodium nitroprusside (Sigma-Aldrich), and 1 mM GTP (Sigma-Aldrich) for 10 min at 37 °C. The reaction was terminated with 900 μL of ice-cold 0.1 N HCl (Sigma-Aldrich) and then placed in a 90 °C water bath for 3 min to denature the protein. Samples were centrifuged for 15 min at 2000× *g* to precipitate protein. The supernatant was collected, dried in a speed vacuum, and resuspended in cGMP EIA buffer (Cayman Chemical, Ann Arbor, MI, USA). Samples were diluted 1:2 in EIA buffer for optimal calculation. Absorbance was detected at 405 nm using an iMark automated plate reader (BioRad, Hercules, CA, USA).

### 2.8. Statistics

Outliers were excluded using Grubb’s test prior to analysis. Data are presented as means ± SEM, and experimental groups were compared using Prism 8 software (GraphPad, San Diego, CA, USA) by two-way ANOVA with multiple comparisons, unless otherwise specified, to assess genotype and hyperoxia exposure effects and their interaction. Statistical significance was set at *p* < 0.05.

## 3. Results

### 3.1. Combined SOD3 KO and Hyperoxia Exposure Does Not Cause More Alveolar Simplification Than SOD3 KO or Hyperoxia Alone

Mean alveolar area for the control group (WT-RA) was 1055 ± 41 μm^2^. SOD3 KO and hyperoxia exposure independently induced a significant increase in alveolar area, which was 1863 ± 242 μm^2^ for the KO-RA group and 2932 ± 270 μm^2^ for the WT-O_2_ group (Figure 1A). The combination of SOD3 KO and hyperoxia exposure (KO-O_2_ group) did not induce further alveolar simplification (2364 ± 134 μm^2^). Representative images are shown in Figure 1B.

### 3.2. Combined SOD3 KO and Hyperoxia Exposure Does Not Cause More Microvascular Remodeling Than SOD3 KO or Hyperoxia Alone

Pulmonary vascular MWT for WT-RA was 0.31 ± 0.006. SOD3 KO and hyperoxia exposure independently induced a significant increase in MWT, which was 0.43 ± 0.03 for the KO-RA group and 0.42 ± 0.03 for the WT-O_2_ group (Figure 2A). The combination of genotype and hyperoxia exposure did not induce a further increase in MWT (0.39 ± 0.03). Representative images are shown in Figure 2B. Mean vessel density as quantified by vWF-stained vessels per 10× field was 6.5 ± 0.6 in WT-RA. KO-RA and WT-O2 both had decreased vessel density (4.1 ± 0.6 and 3.5 ± 0.4, respectively; Figure 2C) compared to WT-RA. KO-O_2_ mice showed no additional reduction in vascular density (Figure 2C). Representative phase-contrast and fluorescent images are shown in Figure 2D.

### 3.3. DNA Oxidation Was Not Affected by Genotype

There was an overall increase in 8-OHdG immunofluorescent intensity in hyperoxia-exposed mice; however, post hoc Tukey’s test for multiple comparisons did not reach statistical significance (Figure 3A). There was no effect of genotype on DNA oxidation. Representative phase-contrast, fluorescent, and overlaid images are shown in Figure 3B.

### 3.4. SOD3 Protein Was Absent in KO Mice, but Genotype Did Not Affect Other Antioxidant Protein Expression

There was no SOD3 protein detected in the KO mice, along with no effect of O_2_ exposure on SOD3 expression. In addition to SOD3, SOD1, SOD2, GPX-1, and catalase expression was assayed by Western blot (Figure 4A). Hyperoxia exposure increased SOD2 expression in both WT and KO mice (1.0 ± 0.1-fold vs. 2.6 ± 0.2-fold in WT; 0.9 ± 0.1-fold vs. 1.9 ± 0.3-fold in KO), as described previously [40]. Hyperoxia exposure also decreased catalase expression in WT mice (1.0 ± 0.03-fold vs. 0.72 ± 0.06-fold). However, there was no effect of genotype on measured antioxidant enzyme expression between groups. Representative blots are shown in Figure 4B.

### 3.5. Combined SOD3 KO and Hyperoxia Increased Total SOD Activity

KO-O_2_ mice had increased total SOD activity when compared to WT-O_2_ mice (77.1 ± 4.2 vs. 60.0 ± 4.5 mU SOD activity/μg protein; Figure 5). Comparison of KO-RA and KO-O_2_ mice did not reach statistical significance (63.5 ± 3.8 vs. 77.1 ± 4.2 mU SOD activity/μg protein, *p* = 0.06).

### 3.6. KO-O2 Mice Increased VEGF and Decreased eNOS in Response to Hyperoxia

KO-O_2_ mice had increased VEGF expression when compared to WT-O_2_ mice (2.7 ± 0.4-fold vs. 1.5 ± 0.1-fold; Figure 6A). In order to evaluate downstream effects on the NO-signaling pathway, we assessed eNOS and sGC protein subunit expression and sGC function (Figure 6B–D). eNOS expression was decreased in O_2_-exposed KO mice but not WT mice. sGC-α and -β expression were both significantly reduced in the O_2_ groups (sGC-α not shown), but there was no genotype effect. Representative blots for VEGF, eNOS, and sGC-β are shown. Despite protein expression differences, there were no significant group differences in measured sGC activity (Figure 6D).

## 4. Discussion

SOD3 has been identified as an antioxidant enzyme whose excess is protective of hyperoxic lung injury and whose absence induces histologic changes similar to those seen in hyperoxic lung injury [19,20,23]. However, the precise function of SOD3 and how this function is affected by hyperoxia remains understudied in the neonatal lung. In the present study, neonatal SOD3 KO mice survive prolonged hyperoxia without exacerbation of alveolar simplification or vascular remodeling that occurs in hyperoxia-exposed WT mice or KO mice in RA ( Figure 1; Figure 2). These results are surprising, as both hyperoxia exposure and SOD3 KO alone induce significant damage to alveoli and pulmonary vasculature.

One possibility for these findings is that SOD3 KO induces changes to alveoli and pulmonary vasculature through local effects on redox state in the extracellular space, while hyperoxia exposure induces changes through mitochondrial reactive oxygen species production [41]. We found overall increased nuclear DNA oxidation in hyperoxia-exposed mice but did not find any genotype differences (Figure 3). The lack of increased DNA oxidation in KO mice could mean that local changes in redox state in SOD3 KO are alone sufficient for the phenotypic changes seen in more global “oxidative stress”. This hypothesis is supported by the fact that a delocalizing mutation that converts arginine to glycine (R213G) allows SOD3 to retain enzymatic function, but induces similar morphologic changes in the pulmonary vasculature [42].

If global hyperoxia exposure and SOD3 KO induce similar phenotypic changes through different mechanisms (local extracellular redox state vs. mitochondrial ROI), then why did the dual hit KO-O_2_ mice not exhibit a worse phenotype? We hypothesized that these mice adapt through upregulation of alternative antioxidants. While we found no differences in measured SOD protein expression (Figure 4), our results demonstrate increased total SOD activity in KO-O_2_ mice when compared to WT-O_2_ mice (Figure 5). One possibility for this discrepancy between protein expression and activity is post-translational modification. Numerous post-translational modifications to SODs, including by reactive oxygen and nitrogen species (such as ONOO^−^), have been shown to inhibit SOD1 and SOD2, although limited study has been done on SOD3 [43]. Another hypothesis for the observed lack of phenotypic differences between WT and SOD3 KO mice exposed to hyperoxia is that SOD3 KO and oxygen exposure cause phenotypic changes through a common pathway that is maximally affected by SOD3 KO or O_2_ exposure. The phenotype of KO-RA mice suggests some degree of baseline injury that may activate the transcription factor nuclear factor erythroid 2-related factor 2 (NRF2). NRF2 is a redox-sensitive regulator of myriad antioxidant genes, including SOD3 [44,45]. It is not known, however, whether SOD3 KO activates NRF2, which would confer resistance to further oxidative stress in the SOD3 KO mice. Further study is needed to investigate these possibilities, as well as the importance of post-translational modifications of SOD enzymes on hyperoxic responses in neonatal lungs.

We initially hypothesized that SOD3 protects lung development through VEGF-mediated effects on NO signaling. In a study using 7 days of postnatal exposure at 95% FiO_2_, Perveen et al. found decreased pulmonary VEGF-A protein and vascular endothelial progenitor cell population density in neonatal O_2_-exposed mice [24]. In the same model, SOD3 overexpression protected against these hyperoxic changes. Interestingly, we did not find decreased VEGF-A protein expression in WT mice after O_2_ exposure in our model but did see significantly increased VEGF-A protein in KO mice after O_2_ exposure (Figure 6A). In addition, our KO mice demonstrated decreased eNOS expression after O_2_ exposure that was not seen in WT mice (Figure 6B).

Indeed, eNOS expression has been shown to be decreased in a lamb model of persistent pulmonary hypertension of the newborn (PPHN) only after O_2_-exposure but not in RA, and eNOS expression was restored in these PPHN-O_2_ lambs with exogenous SOD administration [28]. Lastly, decreased eNOS expression in the KO-O_2_ mice compared to KO-RA animals was not associated with altered sGC activity despite reduced sGC expression (Figure 6B–D). Contrary to our initial hypothesis, this would suggest that NO-independent pathways are responsible for the phenotypic differences between WT and KO mice and for the adaptation of the KO-O2 mice to the dual hit. NO-independent pathways stimulated by VEGF might contribute to relative attenuation of hyperoxic lung injury in our KO mice, and this deserves further study [46,47].

There are several limitations to the present study. Firstly, as it relates to animal models of bronchopulmonary dysplasia, the study was designed to be narrow in scope and assess only the effect of hyperoxia exposure and a single antioxidant enzyme, SOD3. BPD is a chronic illness that results from prematurity combined with prenatal exposures, inflammation, oxygen exposure, mechanical lung injury, genetic predisposition, and nutritional status, among other factors. In addition, neonatal hyperoxic lung injury involves far broader and more complex mechanisms than can be elucidated in a single gene knockout.

Secondly, we used a global SOD3 knockout rather than a tissue- or cell-specific knockout. SOD3 is expressed in vascular smooth muscle throughout the body and has been used to model disease processes in other organ systems, including the kidneys [48], eyes [49], and ischemia/reperfusion [50]. However, SOD3 is found in highest content per gram tissue in the vasculature of the lungs, and, despite global knockout, these mice otherwise have a normal lifespan [20]. In addition, we demonstrated decreased eNOS expression in KO-O_2_ mice, but did not assess eNOS activity, which might be affected by factors unrelated to protein expression [51]. Lastly, enzyme activity assays can be limited in their generalizability to in vivo function. Features such as substrate and cofactor concentrations, post-translational modifications, and compartmentalization can all affect enzymatic function in ways that we did not assess.

## 5. Conclusions

We demonstrated that SOD3 KO mice survive prolonged hyperoxia without exacerbation of HLI phenotype. This survival is associated with increased overall SOD activity and increased VEGF expression. Further study is needed to assess whether these mice survive the dual hit due to a common pathway affected by SOD3 KO and hyperoxia exposure or by adaptation through upregulation of SOD activity and/or NO-independent VEGF signaling.

## Figures and Tables

**Figure 1 antioxidants-10-01236-f001:**
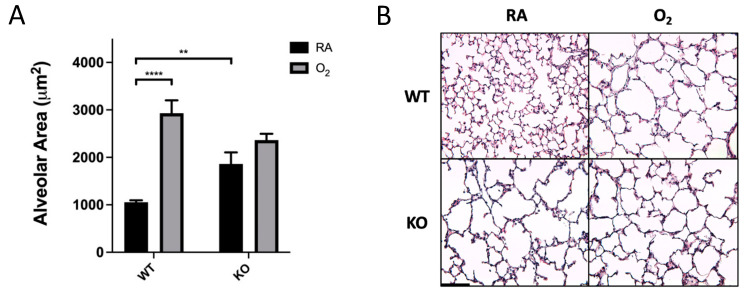
Hyperoxia exposure and SOD3 KO alone induced alveolar simplification, but there was no additive effect of genotype and exposure. (**A**) WT and KO mice were exposed to 14 days of either 75% O_2_ or RA prior to analysis. Lungs were sectioned, H&E-stained, and analyzed at 20× magnification. Alveolar area was significantly higher in O_2_-exposed mice in the WT genotype but not the KO genotype. Data are presented as ± means SEM; ** *p* < 0.01, **** *p* < 0.0001; *n* = 8–15 per group. (**B**) Representative (20×) images of H&E-stained P14 mouse lung. Scale bar is 100 µm.

**Figure 2 antioxidants-10-01236-f002:**
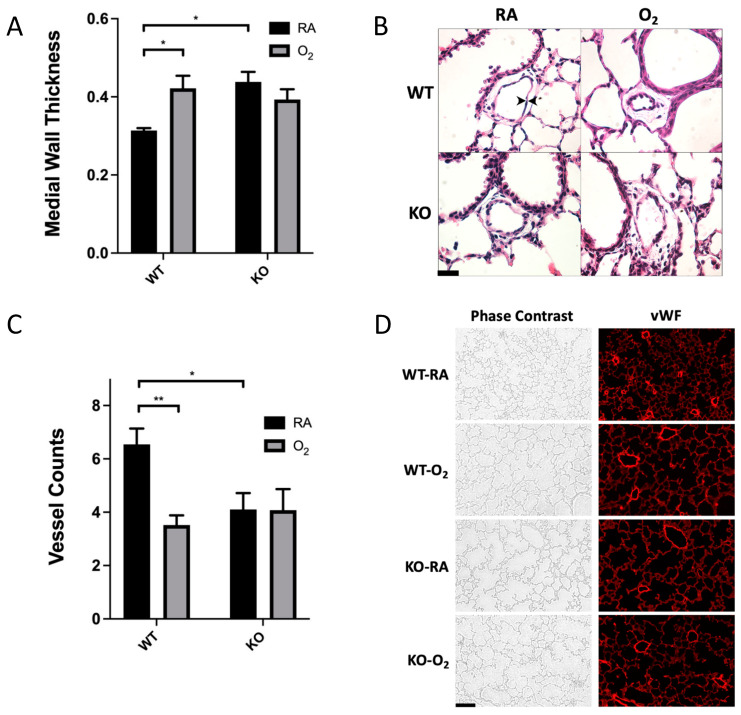
Hyperoxia exposure and SOD3 KO alone induced microvascular remodeling, but there was no additive effect of genotype and exposure. (**A**) Lung sections were analyzed at 40× magnification. Cross-sections of peri-bronchiolar arterioles were identified for analysis. Medial wall thickness was significantly higher in O_2_-exposed mice in the WT genotype but not the KO genotype; *n* = 5–7. (**B**) Representative high-power (40×) images of H&E-stained cross-sections of peribronchiolar pulmonary arterioles. Arrowheads indicate the medial layer. Scale bar is 20 µm. (**C**) Lung sections were stained with vWF antibody to highlight the vascular endothelium. Vascular density was analyzed by counting resistance pulmonary blood vessels (<100 µm diameter) per low-power (10×) field. Vessel counts were significantly lower in O_2_-exposed mice compared to RA in the WT genotype but not the KO genotype. (**D**) Representative low-power (10×) images of phase-contrast and immunofluorescent lung are presented. Scale bar is 100 µm. Data are presented as means ± SEM; * *p* < 0.05, ** *p* < 0.01.

**Figure 3 antioxidants-10-01236-f003:**
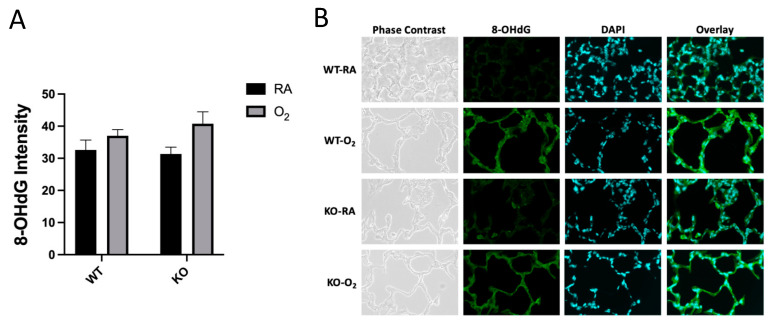
SOD3 KO did not affect DNA oxidation. (**A**) Lung sections were stained for 8-OHdG and counterstained with DAPI. Nuclei were identified on the basis of DAPI fluorescent intensity, and these regions were analyzed for 8-OHdG fluorescent intensity to avoid analyzing background autofluorescence. There was an overall increase in 8-OHdG fluorescent intensity in hyperoxia-exposed mice by two-way ANOVA (*p* < 0.05); however, post hoc individual group comparisons did not reach statistical significance. (**B**) Representative images of phase-contrast, 8-OHdG immunofluorescent (green), and DAPI-stained (blue) lungs are presented. An overlay of the 8-OHdG and DAPI images is shown in the far-right column.

**Figure 4 antioxidants-10-01236-f004:**
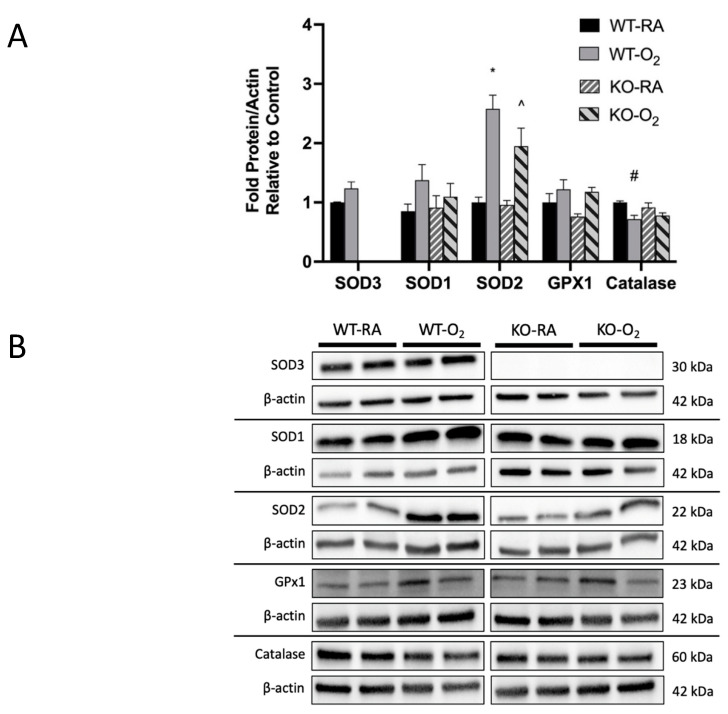
SOD3 protein was absent in KO mice, but other antioxidant enzyme expression was not affected by genotype. (**A**) Western blot analysis was used to evaluate lung protein expression of SOD3, SOD1, SOD2, GPX-1, and catalase in WT and KO mice exposed to either RA or O_2_ for 14 days. Protein expression was normalized to β-actin. No SOD3 protein was detected in KO mice, and O_2_ exposure did not affect SOD3 expression in WT mice. O_2_ exposure was associated with increased SOD2 expression in both genotypes (as previously described [1]) and decreased catalase expression in WT mice. Data are presented as means ± SEM; * *p* <0.0001 vs. WT RA, ^ *p* < 0.05 vs. WT RA, # *p* < 0.01 vs. KO RA, by two-way ANOVA with multiple comparisons; *n* = 4–6 per group. (**B**) Representative blot images are presented.

**Figure 5 antioxidants-10-01236-f005:**
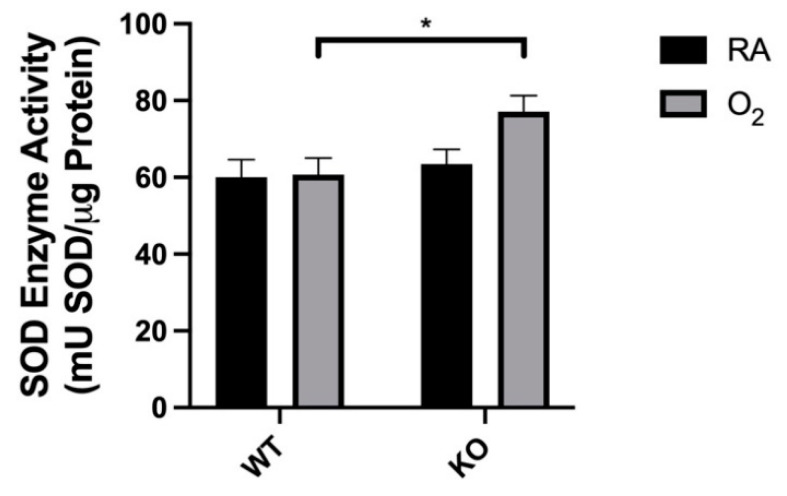
Total SOD activity was increased in KO-O_2_ mice. SOD activity was measured as the rate of elimination of superoxide compared to standard concentrations of SOD enzyme. Whole-lung SOD enzyme activity was increased only in KO-O_2_ mice. Data are presented as means ± SEM; * *p* < 0.05 vs. WT O_2_; *n* = 6–9 per group.

**Figure 6 antioxidants-10-01236-f006:**
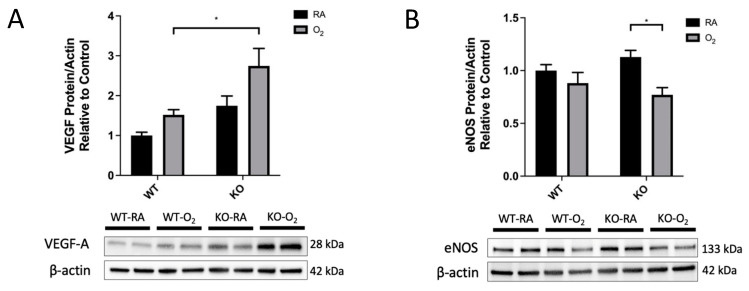
VEGF was increased and eNOS was decreased in combined SOD3 KO and hyperoxia. Western blot analysis was used to evaluate lung protein expression of VEGF-A, eNOS, and sGC-β1 in WT and KO mice exposed to either RA or O_2_ for 14 days. Protein expression was normalized to β-actin. Representative blots are presented. (**A**) VEGF-A expression was significantly increased by the combination of O_2_ and SOD3 KO. (**B**) eNOS expression was significantly decreased by the combination of O_2_ and SOD3 KO. (**C**) sGC-β1 expression was decreased in both genotypes following O_2_ exposure, but there was no genotype effect. sGC-α protein was also measured and had an identical expression pattern (data not shown). (**D**) sGC activity was measured in whole-lung lysates and was not affected by O_2_ exposure or genotype. Data are presented as means ± SEM; * *p* < 0.05, **** *p* < 0.0001 vs. WT RA; *n* = 4–8 per group.

## Data Availability

A supplemental figure is available on FigShare at http://doi.org/10.6084/m9.figshare.15109872. All other data are contained within the article.

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
