# Peer review of "Neonatal Extracellular Superoxide Dismutase Knockout Mice Increase Total Superoxide Dismutase Activity and VEGF Expression after Chronic Hyperoxia"

_antioxidants, 2021, doi:10.3390/antiox10081236_

Round 1

Reviewer 1 Report

In this manuscript, mouse model of neonatal hyperoxic lung injury and SOD3 knockout (KO) mice to evaluate its function during chronic hyperoxia exposure. The results are solid and manuscript is well-documented and organized.

Comments:

  1. In Fig. 6, several bands of representative blots are cutted. The authors should revised this problem.
  2. A Graphic abstract should be provided  for reader to well understand. 

Author Response

Response to Reviewer 1

Manuscript Title: Neonatal Extracellular Superoxide Dismutase knockout mice increase total superoxide dismutase activity and VEGF expression after chronic hyperoxia

Comments and Suggestions for Authors:

In this manuscript, mouse model of neonatal hyperoxic lung injury and SOD3 knockout (KO) mice to evaluate its function during chronic hyperoxia exposure. The results are solid and manuscript is well-documented and organized.

We appreciate this synopsis of the manuscript and feedback. It is clear the reviewer has carefully evaluated the work presented and is able to summarize it accurately and concisely. We have taken into account the suggestions and a point-by-point response is available below.

Comments:

  1. In Fig. 6, several bands of representative blots are cut. The authors should revise this problem.

Figure 6 has been revised to include blot images with clear white borders.

  1. A Graphic abstract should be provided for reader to well understand.

We have modified our Graphical Abstract to more clearly demonstrate the main findings of the research. Please see attachment.

Reviewer 2 Report

The topic of hyperoxia in neonates and mechanisms that lead to BPD is of great interest.  However, this study is limited by considering only a single anti-oxidant. It is simplistic to think that a single antioxidant gene would be responsible for a complex phenotype such as BPD, especially when upstream regulators of numerous antioxidant genes are known.  That being said, it is important to know that SOD3 is probably not a great therapeutic target.

The manuscript could be improved in several ways:

  1. Numerical results for mean alveolar area and MWT should be included in the text for KO-O2 mice along with the other three groups.
  2. mRNA levels for antioxidants, eNOS, etc. in addition to Western blots for protein would be informative. Suggest looking for changes in expression of other antioxidants using RT-PCR as well.
  3. Several hypotheses are put forth in the Discussion to explain the lack of enhanced effects in the KO-O2 group.  A major consideration that is missing is the role of Nrf2 in regulating gene expression of SOD and many other antioxidants. Perhaps an explanation is upstream instead of downstream from SOD3.
  4. Statistics: The Grubb's test for outliers is only appropriate for univariate data. Was the distribution normal?
  5. Concern that the KO on a C57BL/6J was backcrossed to C57BL/6N (There are many genotypic and phenotypic differences between these strains).  How many generations of backcross?  Did heterozygous (SOD3+/-) mice show a phenotype?

Author Response

Please see attached file for reviewer 2 response.

Reviewer 3 Report

The paper describes the effect of SOD3 knockout on the levels of oxidative stress-related enzymes and the lung tissue structure in neonatal mice.

The most evident finding is given in the title.

The subject matter touches on a very sound issue of lung damage during neonatal therapy.

The experimental design is simple and transparent. The research focuses on the examination of the lung tissue structure in normoxia and hyperoxia in mice wt and double SOD3 knockouts. The results are then related to the levels of other key oxidative stress-related enzymes. Also, the DNA oxidative damage is examined, as an indicator of resulting oxidative damage in the tissue.

In the opinion of the reviewer, further investigations on the oxidative stress markers like thiol levels, RNS, ROS (superoxide), and cellular redox potential assays could be performed, probably giving a more precise view. Also, some phosphorylation tests might shed some more light on the cellular pathway responses. Apoptosis/necrosis marker levels study might also add up to a better understanding of the observed effects. increased total SOD and VEGF expression point to a higher level of oxidative stress in knockout/hyperoxia mice. It would be good for the authors and readers to address these issues in the next paper.

The paper in part elucidates the SOD3-related regulatory processes in hyperoxia, although the main questions remain open. Nevertheless, the soundness of the research subject matter, the elegant design, clarity, and relevant discussion provided by the authors make this publication interesting for the readers.

My only major remark is that in my opinion the subject of research tends slightly out of the journal scope, but I would like to leave the decision on that matter to the editor.

A typo in the text: eNOS expression was decreased O2-exposed KO - the ‘in’ is missing.

Author Response

Please see attached file for reviewer 3 response

Round 2

Reviewer 1 Report

The manuscript is well revised according to the reviewer's suggestions and ready to publication.

Reviewer 2 Report

The revised manuscript now acknowledges additional limitations of the study and questions and concerns were adequately answered.